# Facile Preparation of Irradiated Poly(vinyl alcohol)/Cellulose Nanofiber Hydrogels with Ultrahigh Mechanical Properties for Artificial Joint Cartilage

**DOI:** 10.3390/ma17164125

**Published:** 2024-08-20

**Authors:** Yang Chen, Mingcheng Yang, Weiwei Zhang, Wenhui Guo, Xiuqiang Zhang, Benshang Zhang

**Affiliations:** 1Institute of Isotope, Henan Academy of Sciences Co., Ltd., Zhengzhou 450015, China; chenyang@hnas.ac.cn (Y.C.); guowenhui@hnas.ac.cn (W.G.); zhangbenshang@hnas.ac.cn (B.Z.); 2Henan Radiation New Materials Engineering Technology Research Center, Zhengzhou 450015, China; 3Key Biomass Energy Laboratory of Henan Province, Zhengzhou 450008, China; zhangww@hnas.ac.cn (W.Z.); zhangxiuqiang@hnas.ac.cn (X.Z.)

**Keywords:** PVA/CNF hydrogel, γ-ray irradiation, in situ annealing, mechanical strength, artificial joint cartilage

## Abstract

In this study, Poly(vinyl alcohol)/cellulose nanofiber (PVA/CNF) hydrogels have been successfully prepared using γ-ray irradiation, annealing, and rehydration processes. In addition, the effects of CNF content and annealing methods on the hydrogel properties, including gel fraction, micromorphology, crystallinity, swelling behavior, and tensile and friction properties, are investigated. Consequently, the results show that at an absorbed dose of 30 kGy, the increase in CNF content increases the gel fraction, tensile strength, and elongation at break of irradiated PVA/CNF hydrogels, but decreases the water absorption. In addition, the cross-linking density of the PVA/CNF hydrogels is significantly increased at an annealing temperature of 80 °C, which leads to the transition of the cross-sectional micromorphology from porous networks to smooth planes. For the PVA/CNF hydrogel with a CNF content of 0.6%, the crystallinity increases from 19.9% to 25.8% after tensile annealing of 30% compared to the original composite hydrogel. The tensile strength is substantially increased from 65.5 kPa to 21.2 MPa, and the modulus of elasticity reaches 4.2 MPa. Furthermore, it shows an extremely low coefficient of friction (0.075), which suggests that it has the potential to be applied as a material for artificial joint cartilage.

## 1. Introduction

Articular cartilage injury poses a major threat to human health, so effective repair of articular cartilage injury has become an urgent problem [1,2,3]. Traditional surgical treatment has certain limitations and cannot achieve a long-term and satisfactory curative effect. Tissue engineering artificial cartilage technology is expected to play an important role in the field of cartilage treatment. The main materials currently used for artificial cartilage tissue engineering include bioceramics [4,5], titanium alloys [6,7], and composite hydrogels [8]. Hydrogels are viscoelastic cross-linked network polymers capable of adsorbing large amounts of water while maintaining network integrity. Owing to their high water absorption, moisture retention, and good biocompatibility, hydrogels have been widely used in biomedical applications such as artificial joint cartilage material [9,10,11,12]. Among them, Poly(vinyl alcohol) (PVA) hydrogels have been extensively utilized as commercial products in biomedical applications such as contact lenses, wound dressings, and implantable medical materials due to their bioadhesion, nontoxicity, biocompatibility, and high chemical resistance [13,14,15,16].

As a natural material, cellulose nanofibers (CNFs) have a variety of properties different from those of conventional materials, including high specific surface area, rheological properties, liquid crystal behavior, mechanical enhancement, high surface chemical reactivity, biocompatibility, biodegradability, and nontoxicity. Therefore, CNF has been used in the biomedical field [17,18] to enhance the mechanical properties of PVA hydrogels in tissue engineering [19]. This enhancement is usually achieved by hydroxyl hydrogen bonding between CNF and PVA during hydrogel preparation. Tummala et al. reported that the tensile strength of CNF/PVA was nearly three times higher compared to pristine PVA hydrogels [20]. In another study, PVA/CNF hydrogels were prepared by the freeze-drying method using DSMO and water as solvents. It was shown that CNF concentration affects the pore size and compressive strength of hydrogels [21]. In general, CNF/PVA hydrogels formed through hydrogen bonding are usually thermally unstable.

Irradiation is a clean and convenient method for preparing and modifying hydrogels. When ionizing irradiation interacts with polymers, ionization, excitation, ion neutralization, and energy absorption can generate free radicals, while radiation cross-linking can form stable covalent bonds [22,23]. This method is simple, reliable, and allows for the formation and sterilization of hydrogels in a single step, without the need for a chemical initiator.

However, an articular cartilage hydrogel must exhibit three essential characteristics: optimal mechanical properties, biomimetic cartilage microstructure, and robust bioactivity. Irradiated PVA hydrogels are too fragile for artificial joint cartilage materials. As measured by inflation tests and confirmed by finite element modeling, CXL treatment significantly increased the overall stiffness of pig corneas [24]. Improving cross-linking is a feasible way to overcome this problem; annealing began to be used as a means of reinforcing hydrogels. Annealing treatment enhances the mechanical properties of PVA hydrogels by promoting crystalline structure formation and reducing water content. This results in a significant increase in both tensile and compressive strengths [25,26,27].

In this study, a facile strategy for the preparation of PVA/CNF double-cross-linked composite hydrogels was proposed, which consisted of γ-ray-induced chemical cross-linking and physical cross-linking between PVA and CNF. The PVA/CNF hydrogels were first pre-prepared by γ-ray irradiation and then annealed at 80 °C after fixing the ends of the hydrogels. Notably, the stresses generated by the hydrogel due to water loss and shrinkage were used to orient and tightly align the molecular chains of the PVA and the CNFs, resulting in the formation of strong hydrogen bonding interactions. Finally, the double-cross-linked hydrogel was obtained by immersion in deionized water. As a result, the obtained composite hydrogels with double-cross-linked networks showed significant improvement in tensile properties compared to pure irradiated PVA hydrogels without annealing treatment. Furthermore, the composite hydrogels also possessed superb lubrication and load-bearing capacity, indicating the potential applications in artificial joint cartilage.

## 2. Materials and Methods

### 2.1. Materials

PVA (Mw 74,800–79,200 g/mol, 99% hydrolyzed, AR grade) was purchased from Tianjin Damao Chemical Reagent Factory, Tianjin, China. TEMPO-oxidized CNF solution (1.35 wt%) was purchased from Tianjin Mujingling Biotechnology Co., Ltd., Tianjin, China. Phosphate buffered solution (PBS) was purchased from Macklin Co., Ltd., Shanghai, China. All chemicals were of analytical reagent grade and were used as received. Deionized water was used throughout all experiments.

### 2.2. Radiation Preparation of PVA/CNF Hydrogels

The fabrication process of PVA/CNF hydrogels is shown in Figure 1a. The CNF solution was homogenized three times using a high-pressure homogenizer (M-110P, Microfluidics, Westwood, MA, USA) to improve dispersion stability. Then, it was diluted with deionized water to prepare CNF dispersions with varying concentrations of 0, 0.3, 0.6, and 0.9 wt%. Subsequently, 9 g of PVA was added to 91 g of each CNF dispersion. The mixture was stirred at 95 °C for 4 h, degassed at 50 °C for 2 h, and then poured into a silicone mold. Finally, the hydrogels were cross-linked by a ^60^Co γ-ray source with an absorbed dose of 30 kGy. The resulting hydrogels were labeled according to their CNF content as 0% CNF, 0.3% CNF, 0.6% CNF, and 0.9% CNF.

### 2.3. Preparation of Annealed PVA/CNF Hydrogels

The hydrogels formed by γ-ray irradiation were cut to a size of 50 × 80 mm and then annealed at 80 °C for 6 h until the weight remained constant. As shown in Figure 1b and Table 1, the hydrogels were subjected to three treatments during the annealing process: (a) no further treatment (Annealing), (b) fixation of both ends of the hydrogel (In situ annealing), and (c) the hydrogel was stretched by 30% and subsequently fixed at both ends (30% stretch).

After annealing, the dried hydrogels were rehydrated in deionized water for 24 h to reach swelling equilibrium.

### 2.4. Gel Fraction Determination

Unannealed PVA/CNF hydrogel samples were initially dried until their weight (*W_o_*) remained constant in a thermostatic oven at 50 °C. Subsequently, the dried hydrogel samples were rehydrated in deionized water at 100 °C for 48 h to remove the uncross-linked components. The remaining gel was then dried at 50 °C for 48 h to a constant weight (*W_g_*). Gel fraction (*G*%) was calculated gravimetrically using Equation (1):(1)$$G\%=\frac{W_{g}}{W_{o}}\times 100\%$$
where *W_o_* and *W_g_* represent the weights of the dried sample before and after extraction, respectively.

### 2.5. Fourier Transform Infrared Spectrometer (FTIR) Analysis

FTIR spectroscopy was used to investigate the alterations in the molecular structures of PVA/CNF hydrogels. The samples were subjected to freeze-drying, followed by grinding into a fine powder and subsequent mixing with potassium bromide (KBr) for spectral analysis. The FTIR spectra were obtained over a wavenumber range of 500–4000 cm^−1^ using a Nicolet iS10 FTIR spectrometer (Thermo Fisher Scientific, Waltham, MA, USA).

### 2.6. Micromorphology Characterization

Scanning Electron Microscopy (SEM) images were captured using a JSM-IT800 Schottky Field Emission Scanning Electron Microscope (FESEM) (JEOL, Tokyo, Japan). PVA/CNF hydrogel samples were freeze-dried and then fractured in liquid nitrogen to preserve the micromorphology. The fracture surfaces were sputter-coated with gold for 80 s before SEM examination.

### 2.7. Swelling Test

The swelling degree of PVA/CNF hydrogels in deionized water was investigated by measuring the weight change over time. Hydrogel weight was measured at 0, 1, 3, 5, 6, 12, 15, and 24 h. Prior to weighing, the rehydrated gel was dried on filter paper to remove water on its surface. The swelling degree (*W*%) was calculated using Equation (2):(2)$$W\%=\frac{W_{t}-W_{o}}{W_{o}}\times 100\%$$

*W_t_* represents the weight of the hydrogel sample at swelling equilibrium in deionized water. For unannealed hydrogels, *W_o_* represents the weight measured immediately after radiation cross-linking. For annealed hydrogels, *W_o_* represents the weight of the dried hydrogel after annealing.

### 2.8. Differential Scanning Calorimetry (DSC) Analysis

DSC measurements were performed using a Q2000 DSC apparatus (TA Instruments, New Castle, PA, USA). Approximately 3–5 mg of freeze-dried PVA/CNF hydrogel samples were subjected to a temperature ramp from 50 to 250 °C at a heating rate of 10 °C/min. The experimental procedure was conducted under a nitrogen atmosphere with a flow rate of 100 mL/min. The crystallinity of the samples was determined using the following Equation (3):(3)$$X_{c}=\frac{∆H}{{∆H}_{c}}\times 100\%$$

The enthalpy of fusion (Δ*H*) of the hydrogels was determined by integrating the melting endotherm. The theoretical enthalpy of fusion for 100% crystalline PVA (Δ*H_c_*) is 138.6 J/g [28].

### 2.9. Tensile Properties Test

The tensile properties of PVA/CNF hydrogels were evaluated using a universal testing machine, AGS-X 100N (Shimadzu, Kyoto, Japan). Specimens with a dumbbell shape, measuring 2 mm in thickness and 20 mm in length, were fabricated using a specimen slicer. These specimens were then subjected to tensile testing at a stretching speed of 50 mm/min.

### 2.10. Friction Properties Test

The friction properties of PVA/CNF hydrogels were evaluated using a ball-on-disk tribometer (MGW-02, Lanzhou, China). Hydrogel samples were soaked in deionized water to reach swelling equilibrium before testing. A 5 mm diameter stainless steel ball was used as the counterface and PBS served as the lubricant. The test was conducted at 5 Hz for 9000 cycles under a 10 N load.

## 3. Results and Discussion

### 3.1. Preparation of the PVA/CNF Hydrogels

The preparation of PVA/CNF composite hydrogels is shown in Figure 2. Chemical cross-linking of the PVA/CNF mixtures was induced by irradiating the PVA/CNF mixtures with γ-rays at an absorbed dose of 30 kGy.

Figure 3 shows the variation in gel fraction and CNF concentration of PVA/CNF hydrogels at an absorbed dose of 30 kGy. It was found that the gel contents of the samples were 94.1%, 95.9%, 97.2%, and 97.4%, respectively. Under γ-ray irradiation, PVA aqueous solution mainly absorbs energy from water and produces short-lived reactive species such as hydrogen atoms and hydroxyl radicals. Specifically, hydrogen atoms on the α-position of the hydroxyl group (–CH(OH)–) react preferentially to form tertiary radicals, while hydrogen atoms on the methylene group (–CH_2−_) form secondary radicals. Subsequently, these radicals form a three-dimensional polymer network in the PVA hydrogel through recombination interactions and mutual influence [29]. The higher the CNF content in the hydrogel, the greater the proportion of the gel fraction, which indicates that CNF acts as a cross-linker (cross-linking promoter). Under irradiation, the hydroxyl and methylene (–CH_2−_) groups in CNF also produce free radicals, which promote cross-linking of the PVA molecular chains. The incorporation of CNF enhances the possibility of free radical recombination in PVA, thereby increasing the cross-linking density in the hydrogel [30].

Figure 3 shows the FTIR spectra of the PVA, PVA/CNF, and 30% stretch annealed PVA/CNF hydrogels. The characteristic absorption peaks of PVA around 3430 cm^−1^ (–OH stretching vibration), 2922 cm^−1^ (–CH stretching vibration), 1428 cm^−1^ (–CH_2_ deformation vibration), and 1091 cm^−1^ (C–O stretching vibration) were exhibited in all three FTIR spectra. Compared with PVA hydrogels, –OH absorption peaks of PVA/CNF hydrogels were wider, and their wavenumber gradually shifted to a lower value of 3409 cm^−1^. These differences are caused by the hydrogen bonding interactions between PVA and CNF. The absorption peaks at 1144 cm^−1^ of PVA/CNF hydrogels were more obvious than those of PVA hydrogels, which is attributed to the C–C stretching vibration of PVA and is an indicator of the crystallinity [21].

### 3.2. Morphology of Hydrogels

As shown in Figure 4, the radiation-prepared PVA/CNF hydrogels exhibit excellent transparency both before and after annealing, which is not available for the physiological PVA hydrogels prepared by the freeze-drying method. The SEM image in Figure 4b reveals the reticular structure of the unannealed PVA hydrogel with an apparent pore size of about 85 μm. Figure 4d,f shows the microscopic morphology of the naturally annealed and in situ-annealed hydrogels after swelling, respectively, which indicates that the unannealed hydrogel does not have the typical re-reticulation structure. This transformation is attributed to the significant increase in the physical cross-linking density of the polymer molecular chains during the annealing process. The difference between the surface layer and the hydrogel center in Figure 4f may be due to the different water loss rates during the annealing process. It is worth noting that the annealing temperature of 80 °C exceeds the glass transition temperature of PVA, resulting in rapid evaporation of water from the surface layer. Meanwhile, the molecular chains are aligned along the ends of the tensile stress, resulting in a fine-line surface topography along the stress direction.

### 3.3. Swelling Behavior

The swelling kinetics of PVA/CNF hydrogels in deionized water at 25 °C are shown in Figure 5a. All samples reached swelling equilibrium after 24 h of immersion, indicating that the CNF content exhibited no significant effect on the water absorption rate of PVA hydrogels. Figure 5a shows that the equilibrium swelling ratio decreased with increasing CNF content because CNF enhances the hydrogen bond interaction between the PVA molecular chain and CNF. This strong interaction restricts the movement of the PVA molecular chain, further limiting the swelling behavior of the hydrogels [21].

The swelling degree of annealed PVA/CNF hydrogels is shown in Figure 5b. It could be noticed that the equilibrium swelling of the hydrogels increased with increasing CNF content, which is consistent with the results of the unannealed hydrogels and gel fraction test. The increase in CNF content enhances the cross-linking of the hydrogel and reduces the diffusion space for water molecules in the polymer network, which in turn inhibits the swelling of the hydrogel. Comparing the various annealing methods, natural annealing, 30% stretch annealing, and in situ annealing have progressively less effect on hydrogel swelling. The reason for this is that natural annealing leads to inward shrinkage in six directions, whereas the other two methods limit the shrinkage to four directions by fixing the ends of the hydrogel [31]. As a result, naturally annealed hydrogels exhibit the smallest polymer network space and the lowest swelling degree.

### 3.4. DSC Analysis

The DSC curves of 0.6% CNF hydrogels are shown in Figure 6. There is an obvious heat absorption peak in the range of 220–230 °C which indicates that the hydrogels have a certain degree of crystallinity. Table 2 reveals that both in situ annealing and 30% deformation stretch annealing increased the Tm of the hydrogels by 2.9 °C and 3.2 °C, respectively, indicating that the thermal stability of the hydrogels was improved. Compared with the untreated curve (a), the heat absorption peak of the annealed hydrogel is sharper, signifying a more regular crystal distribution. Table 2 summarizes the degree of crystallinity based on the integrated area of the crystalline peaks, highlighting that the annealing process significantly improved the degree of crystallinity. Specifically, in situ annealing and 30% deformation stretch annealing resulted in an increase in crystallinity of 5.7% and 5.9%, respectively, compared to unannealed hydrogels, while naturally annealed hydrogels showed an increase of 2.7%. This difference is attributed to the shrinkage of the hydrogel due to water evaporation during the annealing process, which leads to an increase in crystallinity due to the reduction in the gap between the PVA and CNF molecular chains and the enhancement of hydrogen bonding. Furthermore, immobilizing the ends of the hydrogel enhances the vertical stress and promotes the alignment and regularity of the molecular chains, thus improving the crystallinity [32,33].

### 3.5. Tensile Property

The effect of CNF content on the tensile properties of PVA/CNF hydrogels without the annealing procedure is schematically shown in Figure 7. As shown, the stress–strain curves of the PVA/CNF hydrogels exhibit the characteristic behavior of superelastic materials, which is in contrast to the behavior observed in pure PVA hydrogels, where a large elongation occurs at low stresses and a gradual stiffening occurs with the increase in the applied stress. Figure 7b shows that the tensile strength and elongation at break of the PVA/CNF hydrogels were improved significantly with increasing CNF content. The tensile strength and the elongation at break of the PVA/CNF hydrogels (0.9% CNF content) were 91.4 kPa and 207.1%, respectively. Notably, these values were 433% and 34% higher than those of the pure PVA hydrogels, respectively.

The large number of hydroxyl and carboxylic groups on the surface of CNFs could form hydrogen bonds with the PVA polymer through a high degree of interaction with the hydroxyl groups of the PVA polymer chains. Meanwhile, the high degree of physical cross-linking could also improve the irradiated cross-linking network. Rigid cellulose nanofibers were used as spacers to provide support for the porous network structure [34]. These factors are the main reasons for the increase in tensile strength and elongation at break of PVA/CNF hydrogels with increasing CNF content.

As shown in Figure 8, the rehydrated PVA/CNF hydrogel with an annealing strain of 30% can easily lift a weight of 1 kg with a width and thickness of only 3.2 mm and 0.4 mm, respectively. Figure 9a shows the tensile stress–strain curves of the PVA/CNF hydrogels (0.6% CNF) after three different annealing methods, and all three curves also show the behavioral characteristics of the superelastic material. The tensile strength and elongation at break of the unannealed PVA/CNF hydrogels were 65.6 kPa and 194.1%, respectively, whereas the tensile strength of the annealed, in situ-annealed, and 30% stretch annealed PVA/CNF hydrogels was 15.3, 17.4, and 21.2 MPa, with an increase in tensile strength and elongation at break by a factor of 320 and 1.7, respectively. Furthermore, among the hydrogels obtained by the three annealing methods, the annealing showed the highest elongation at break and the lowest tensile strength, while the 30% strain method showed the lowest elongation at break and the highest tensile strength. This may be due to the fact that during the annealing process, the annealed hydrogel shrinks inward from all sides and the arrangement of the molecular chains remains disordered. In situ and 30% stretch annealing fixed the ends of the hydrogels, and the direction of stress generated during the annealing process was parallel to it, so that the molecular chains of PVA and CNF were oriented to some extent in the fixed directions. The molecular chains are more regularly and tightly arranged, resulting in a higher degree of crystallinity.

In the tensile tests, the higher crystallinity materials exhibited higher tensile strengths but were more prone to fracture, which is consistent with the highest crystallinity of the hydrogels treated with 30% strain in DSC. Figure 9b–d shows the effects of different CNF contents and annealing methods on the tensile modulus, tensile strength, and elongation at break of PVA/CNF hydrogels, respectively. The pure PVA hydrogel was not able to withstand 30% strain due to its poor strength, resulting in fracture during annealing. As can be seen in Figure 9b,c, the elastic modulus increases gradually with increasing CNF content, which reflects the hardness of the material. However, the tensile strength of the hydrogels treated with 30% stretch annealing decreased at 0.9% CNF content, which may be due to the disruption of the strong hydrogen bonding between the PVA molecular chain. The rigid CNF molecular chain may hinder the rearrangement of the PVA molecular chain under stress, resulting in a decrease in the properties. In Figure 9d, it can be seen that the addition of CNF can improve the elongation at break of the annealed hydrogel, but the elongation at break decreases with the increase in CNF content. Moreover, the tensile strength and elastic modulus of the stretched 30% annealed hydrogels were the highest and elongation at break was the lowest for the same CNF content. During the annealing process, the hydrogels will shrink. Immobilizing the ends of the hydrogel or subjecting it to a 30% stretch results in the alignment of the shrinkage-induced stress parallel to the stretching direction. Consequently, CNF and PVA molecular chains are oriented in the direction of the shrinkage-induced stress, leading to an increase in tensile strength and a decrease in elongation at break. Both the addition of CNF and the application of 30% stretch annealing promote the orientation of the molecular chains to some extent.

### 3.6. Friction Property

Artificial joint materials generate a lot of friction and wear during movement. Therefore, the friction coefficient of the in situ-annealed hydrogel was tested and evaluated in order to bring its tensile strength in line with the standards for cartilage materials, as shown in Figure 10. PBS served as a lubricant in these experiments. As the CNF content increased, the friction coefficients showed a decreasing trend: 0.171 for 0% CNF, 0.101 for 0.3% CNF, 0.085 for 0.6% CNF, and 0.075 for 0.9% CNF, which indicated that the PVA/CNF hydrogels showed better lubrication and load-bearing capacity compared to the pure PVA hydrogels. The friction coefficient of the PVA hydrogel prepared by the freeze–thaw method using DMSO as solvent was 0.12 under a load of 10 N [35]. Similar results were obtained for the PVA/PEG composite gel [36].

It is noteworthy that only pure PVA hydrogels ruptured under a force of 10 N, leading to an increase in the contact area and indentation depth, which resulted in a higher friction factor for PVA hydrogels [37,38]. The abundance of OH− and COOH− on the CNF surface promotes water adsorption and lubrication layer formation, which improves the friction properties of the hydrogel. This improvement is also confirmed by the accelerated water absorption rate of CNF-containing hydrogels observed in the swelling test.

## 4. Conclusions

In this study, PVA/CNF hydrogels have been successfully prepared by γ-ray irradiation, annealing, and rehydration. At an absorbed dose of 30 kGy, higher CNF content resulted in higher gel fraction, tensile strength, and elongation at break of irradiated PVA/CNF hydrogels with lower water absorption. Furthermore, annealing the PVA/CNF hydrogels at 80 °C significantly increased the cross-linking density, leading to a change in the micromorphology from a porous network to a smooth plane. For the PVA/CNF hydrogels with 0.6% CNF content, a significant improvement was observed after 30% stretch annealing at 80 °C compared to the original composite hydrogel. The crystallinity was increased from 19.9% to 25.8%, while the tensile strength was significantly increased from 65.5 kPa to 21.2 MPa, with an elastic modulus of 4.2 MPa. The friction coefficient of PVA/CNF hydrogels decreased with the increase in CNF content.

Overall, the stretching annealed PVA/CNF hydrogels demonstrated enhanced strength and lubrication properties, highlighting their great potential as materials for artificial joint cartilage. The future of artificial cartilage development lies in constructing a biomimetic microenvironment that closely replicates natural cartilage tissue. This requires meticulous engineering of both the physical and functional structures. Integrating multiple materials, incorporating gradient structures, and exploring diverse fabrication techniques offer promising avenues for achieving this complex goal. Stretching annealing is an effective strategy for enhancing the mechanical properties of cross-linked hydrogels. However, the complex physiological and mechanical demands imposed on cartilage present significant challenges in developing artificial cartilage capable of meeting the requirements of load bearing, friction reduction, and long-term durability. Moreover, comprehensive assessments of material stability, biocompatibility, and degradation behavior are essential to ensure the suitability of these materials for implantation in the human body.

## Figures and Tables

**Figure 1 materials-17-04125-f001:**
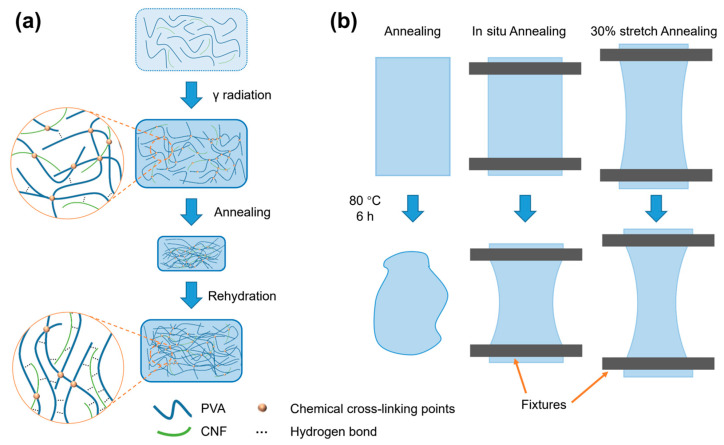
Schematic diagram of (**a**) the PVA/CNF hydrogel preparation and (**b**) Annealing treatment.

**Figure 2 materials-17-04125-f002:**
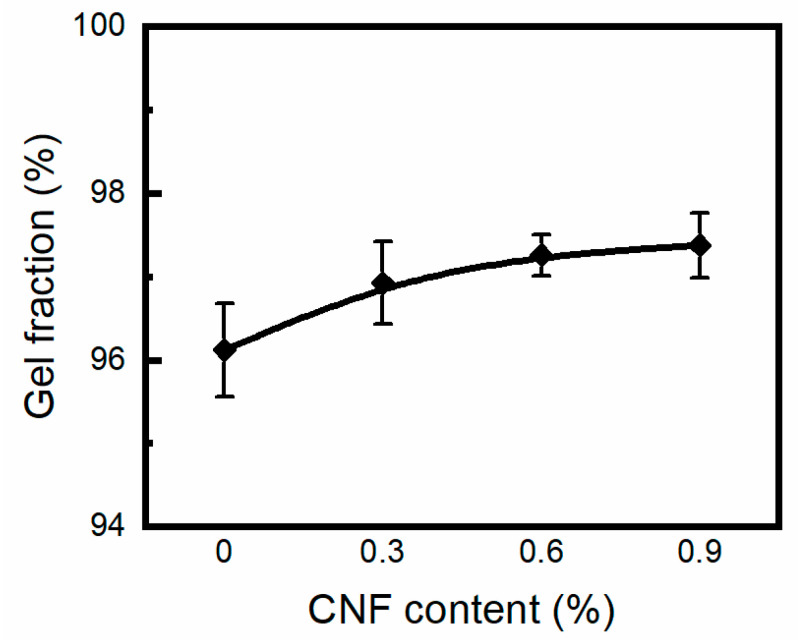
Gel fraction of irradiated PVA/CNF hydrogels.

**Figure 3 materials-17-04125-f003:**
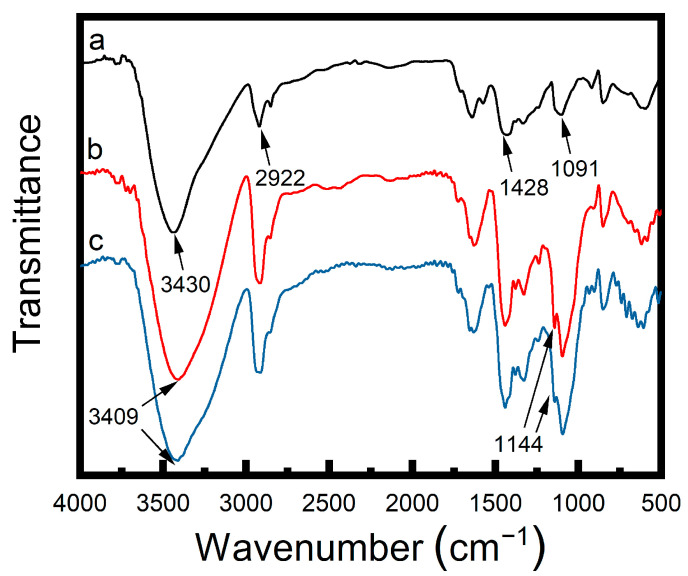
FTIR spectra of the (**a**) PVA hydrogels, (**b**) irradiated PVA/CNF hydrogels (CNF 0.9%), and (**c**) 30% stretch annealed PVA/CNF hydrogels (0.9%).

**Figure 4 materials-17-04125-f004:**
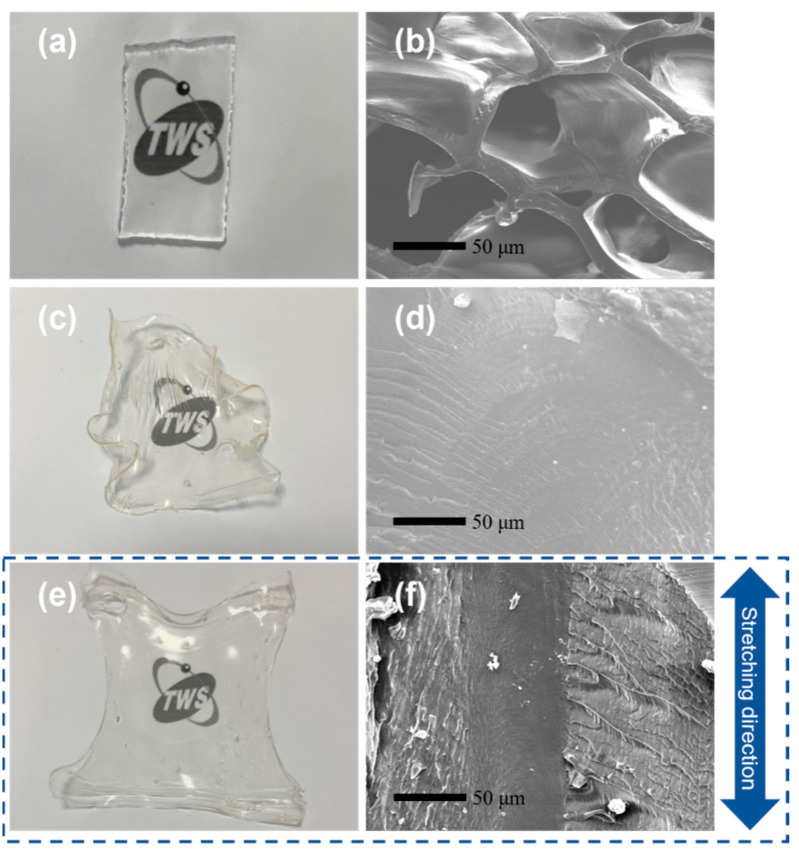
Photos and SEM images of the PVA/CNF hydrogels. (**a**,**b**) Irradiated PVA hydrogels, (**c**,**d**) annealed PVA/CNF hydrogels (0.9% CNF), (**e**,**f**) in situ-annealed PVA/CNF hydrogels (0.6% CNF).

**Figure 5 materials-17-04125-f005:**
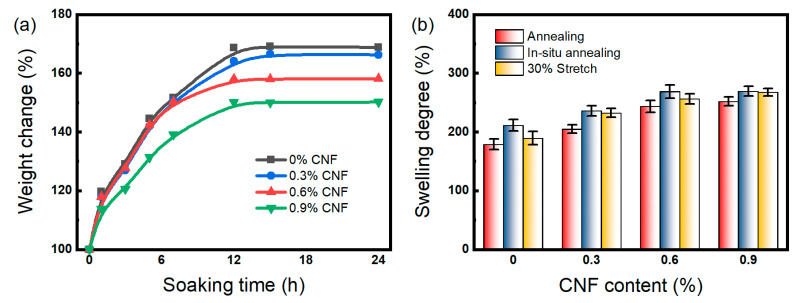
(**a**) Swelling behavior curves of irradiated PVA/CNF hydrogels, (**b**) equilibrium swelling degree of annealed PVA/CNF hydrogels.

**Figure 6 materials-17-04125-f006:**
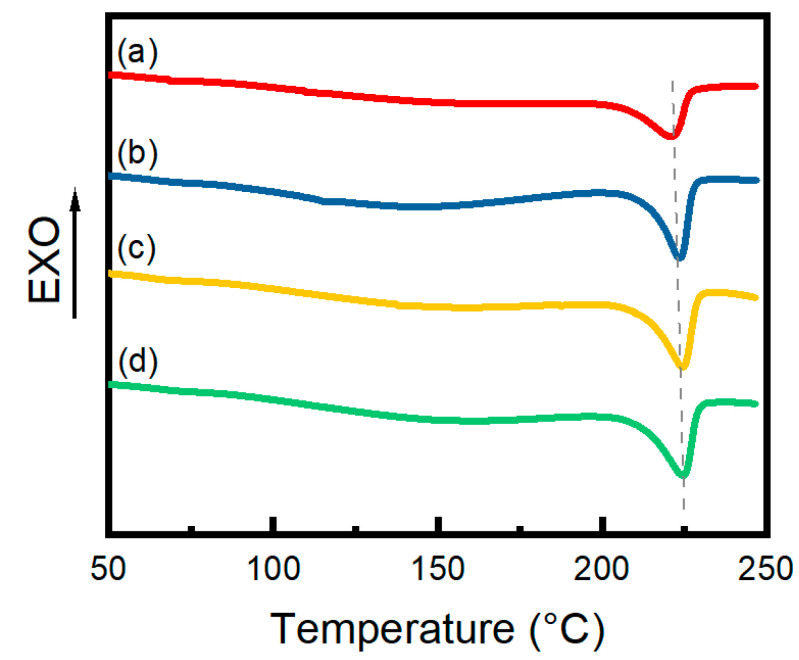
DSC thermograms of 0.6% CNF hydrogels. (**a**) Irradiated hydrogels, (**b**) annealed hydrogels, (**c**) in situ-annealed hydrogels, (**d**) 30% stretch annealed hydrogels.

**Figure 7 materials-17-04125-f007:**
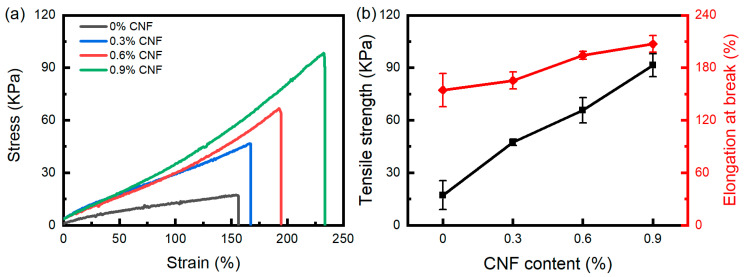
(**a**) Tensile stress–strain curves and (**b**) tensile strength and elongation at break of irradiated PVA/CNF hydrogels.

**Figure 8 materials-17-04125-f008:**
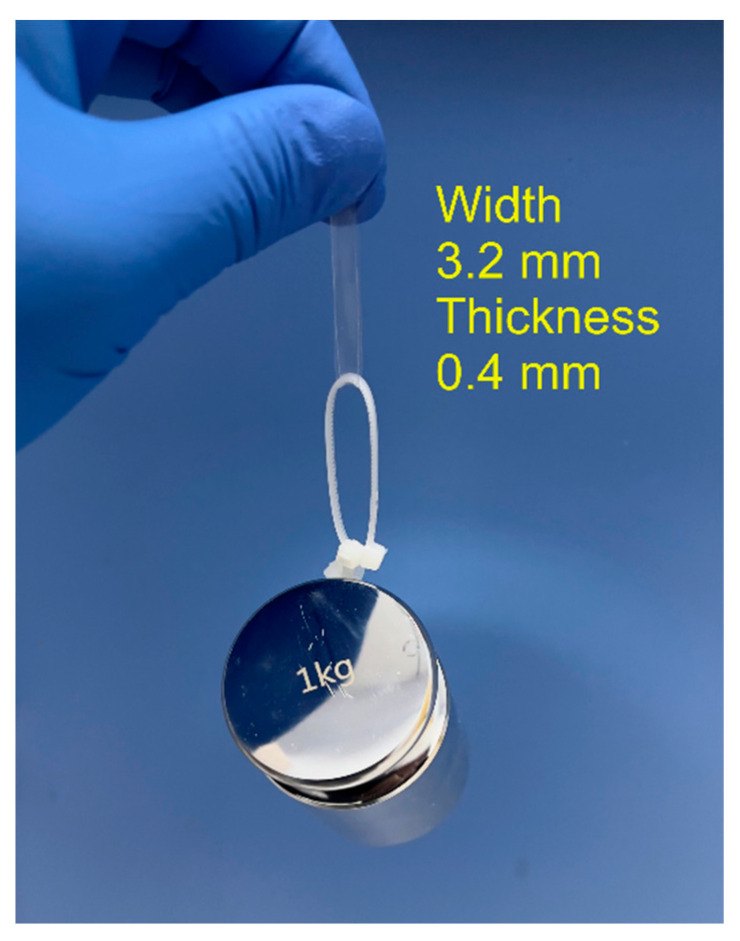
1 kg of weight lifted by the 0.6% CNF with 30% stretch hydrogel.

**Figure 9 materials-17-04125-f009:**
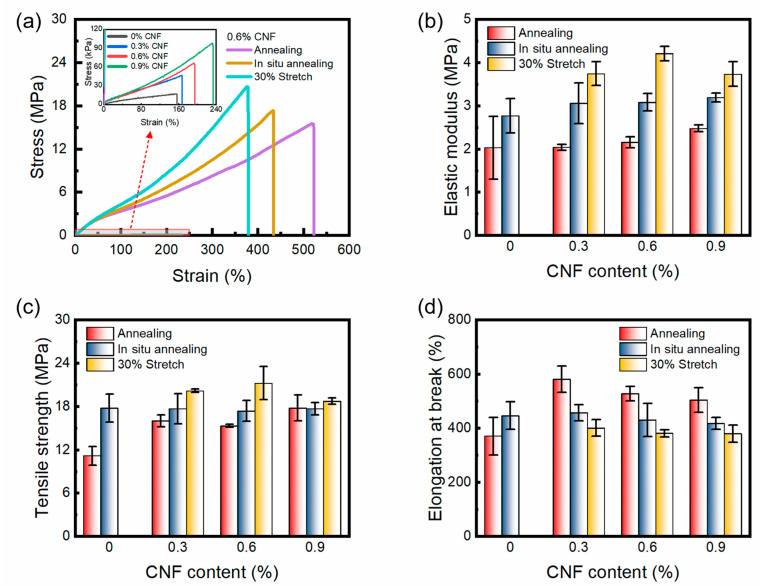
Tensile strength and elongation at break of PVA/CNF hydrogels. (**a**) Stress–strain curves, (**b**) elastic modulus, (**c**) tensile strength, (**d**) elongation at break.

**Figure 10 materials-17-04125-f010:**
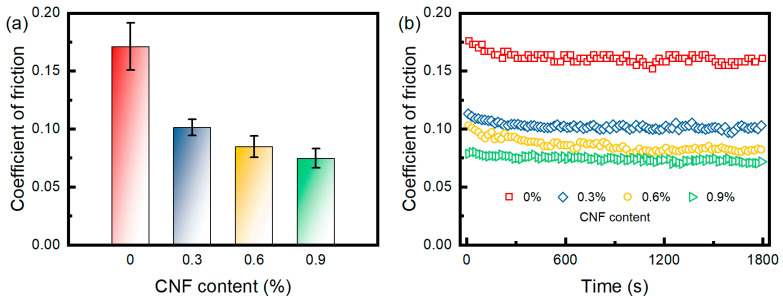
Friction properties of in situ-annealed PVA/CNF hydrogels: (**a**) long time friction test curves, (**b**) coefficient of friction.

**Table 1 materials-17-04125-t001:** PVA/CNF hydrogel composition and annealing methods.

Sample	PVA Content (wt%)	CNF Content (wt%)	Annealing Treatment
1	9	/	/
2	9	/	Annealing
3	9	/	In situ annealing
4	9	0.3	/
5	9	0.3	Annealing
6	9	0.3	In situ annealing
7	9	0.3	30% stretch
8	9	0.6	/
9	9	0.6	Annealing
10	9	0.6	In situ annealing
11	9	0.6	30% stretch
12	9	0.9	/
13	9	0.9	Annealing
14	9	0.9	In situ annealing
15	9	0.9	30% stretch

**Table 2 materials-17-04125-t002:** Crystallinity of PVA/CNF hydrogels.

0.6% CNF Hydrogels	Tm (°C)	Xc (%)
Unannealed	221.5	19.9%
Annealed	223.7	22.6%
In situ-annealed	224.4	25.6%
30% stretch annealed	224.7	25.8%

## Data Availability

The data used in this study are available from the corresponding authors upon reasonable request.

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
