# Peer review of "Facile Preparation of Irradiated Poly(vinyl alcohol)/Cellulose Nanofiber Hydrogels with Ultrahigh Mechanical Properties for Artificial Joint Cartilage"

_materials, 2024, doi:10.3390/ma17164125_

Round 1

Reviewer 1 Report

Comments and Suggestions for Authors

The present article work entitled “Facile Preparation of Irradiated Poly(vinyl alcohol)/cellulose Nanofiber Hydrogels with Ultrahigh Mechanical Properties for Artificial Joint Cartilage” is an article presenting a comparative study of the properties of Poly(vinyl alcohol)/cellulose nanofiber (PVA/CNF) hydrogels with different PVA/CNF ratios. The impact of different annealing methods on the hydrogel properties were tested, including gel fraction, micromorphology, crystallinity, swelling behavior, tensile and friction properties. There are enough and relevant references. The conclusions are consistent with the evidence and arguments presented. The English language should be polished.

Overall, I think that this work can be published in Materials after revision. Some points that can be addressed are mentioned below:

-The manuscript content could be enriched if the authors added some characterization results with techniques like FTIR and XRD.

 -Page 5, Line 173: I guess the authors mean Figure 3b and not 3d?

 - From the Figure 4b, I observe that the swelling degree increases with increasing the CNF content. The authors explain in the text that the swelling ratio decreases with increasing the CNF content. Is it something that I do not get?

 -Page 8, Line 254: Figure 7e is Figure 8, the authors should correct it and take care of the right order/numbering of the figures in the text.

 -In Figure 7a, mention in the text or in the figure caption the inset of Fig.7a.

Comments on the Quality of English Language

the English language could be polished, there are some minor mistakes

Author Response

Comments 1: The manuscript content could be enriched if the authors added some characterization results with techniques like FTIR and XRD.

Response 1: Thank you for your comment. We have included the FTIR spectra in the manuscript as requested. Unfortunately, we do not have an XRD diffractometer and are unable to arrange testing with an external laboratory within the required 7-day timeframe. If such conditions arise later, we will immediately make up for it and publish our results.

Comments 2: Page 5, Line 173: I guess the authors mean Figure 3b and not 3d?

Response 2: Thank you for your comment. The correct figure caption is 3b, the text has been revised.

Comments 3: From the Figure 4b, I observe that the swelling degree increases with increasing the CNF content. The authors explain in the text that the swelling ratio decreases with increasing the CNF content. Is it something that I do not get?

Response 3: Thank you for your comment. Before annealing, the swelling ratio of PVA/CNF hydrogel decreases with increasing the CNF content. But the swelling ratio increases after annealing. We have corrected the text to avoid misunderstanding.

Comments 4: Page 8, Line 254: Figure 7e is Figure 8, the authors should correct it and take care of the right order/numbering of the figures in the text.

Response 4: Thank you for your comment. The correct figure caption is 3b, the text has been revised.

Comments 5: In Figure 7a, mention in the text or in the figure caption the inset of Fig.7a.

Response 5: Thank you for your comment. All the figure captions have been revised in the manuscript.

Response to Comments on the Quality of English Language

Comment 1: the English language could be polished, there are some minor mistakes.

Response 1: Thank you for your suggestions. We revised the whole manuscript carefully to avoid language errors. In addition, we consulted a professional editing service to check the English. We believe that the language is now acceptable for the review process.

Reviewer 2 Report

Comments and Suggestions for Authors

1. The article's main goal is to develop new methods for obtaining versatile and innovative materials for the fabrication of artificial joint cartilage. The Introduction section and the Conclusion (now the final phrase) should present this aspect more clearly.

2. After analyzing the original article, I can confidently say that the methodology is well-presented, the entire article is fluent, and it follows logic.

My observations also refer to the resolution of the figures (starting with 4a, 5, etc.) in order to maximize their quality.

3. What stability studies should develop in future research?

4. Please recheck all the references in order to follow all the requirements for the articles.

Comments on the Quality of English Language

The analysed manuscript requires moderate editing of English language.

Author Response

Comments 1: The article's main goal is to develop new methods for obtaining versatile and innovative materials for the fabrication of artificial joint cartilage. The Introduction section and the Conclusion (now the final phrase) should present this aspect more clearly.

Response 1: Thank you for your comment. The Introduction and Conclusion section have been revised in the manuscript.

Comments 2: After analyzing the original article, I can confidently say that the methodology is well-presented, the entire article is fluent, and it follows logic. My observations also refer to the resolution of the figures (starting with 4a, 5, etc.) in order to maximize their quality.

Response 2: Thank you for your comment. All the figures have been replaced with high resolution ones.

Comments 3: What stability studies should develop in future research?

Response 3: Thank you for your comment. Stability studies will be conducted through structural changes and wear behavior under specific load, pressure and environmental conditions in future research.

Comments 4: Please recheck all the references in order to follow all the requirements for the articles.

Response 4: Thank you for your suggestions. All the references have been revised to follow the requirements.

 Response to Comments on the Quality of English Language

Comment 1: The analysed manuscript requires moderate editing of English language.

Response 1: Thank you for your suggestions. We revised the whole manuscript carefully to avoid language errors. In addition, we consulted a professional editing service to check the English. We believe that the language is now acceptable for the review process.

Reviewer 3 Report

Comments and Suggestions for Authors

The paper deals with sample preparation and material (mechanical) property characterisation of irradiated Poly Vinyl Alcohol – Cellulose Nanofibre Hydrogels, with possible application to artificial joint cartilage tissues.

The subject appears of interest for the scientific community and for possible innovative applications.

However, regarding the current version of the manuscript, some observations can be raised:

- the Introduction section is lacking of references to research works on computational aspects related to the subject (see, e.g., https://doi.org/10.1007/s42558-020-00030-7);

- the second section, Materials and Methods, requires a significant revision, both with respect to the investigated materials and to the adopted methodologies, i.e., the section is expected to be significantly expanded with explanations, descriptions, modelling features and properties, data and figures/schemes;

- the current version of Conclusions is limited to a brief summary of the obtained results; on the contrary, such section is expected to provide a wider elaboration on the obtained results, toward their general interpretation/meaningfulness and innovative aspects.

Comments on the Quality of English Language

The English text and style shall be carefully revised to avoid misprints and to improve the readability (e.g., trying to avoid too short and split sentences).

Author Response

Comments 1: the Introduction section is lacking of references to research works on computational aspects related to the subject (see, e.g., https://doi.org/10.1007/s42558-020-00030-7);

Response 1: Thank you for your comment. The Introduction section has been revised in the manuscript.

Comments 2: the second section, Materials and Methods, requires a significant revision, both with respect to the investigated materials and to the adopted methodologies, i.e., the section is expected to be significantly expanded with explanations, descriptions, modelling features and properties, data and figures/schemes;

Response 2: Thank you for your comment. FTIR spectra have been added as characterization results. But the response time is only 7 days, it is difficult for us to make significant revisions.

Comments 3: the current version of Conclusions is limited to a brief summary of the obtained results; on the contrary, such section is expected to provide a wider elaboration on the obtained results, toward their general interpretation/meaningfulness and innovative aspects.

Response 3: Thank you for your comment. The Conclusions have been revised in the manuscript.

Response to Comments on the Quality of English Language

Point 1: The English text and style shall be carefully revised to avoid misprints and to improve the readability (e.g., trying to avoid too short and split sentences).

Response 1: Thank you for your suggestions. We revised the whole manuscript carefully to avoid language errors. In addition, we consulted a professional editing service to check the English. We believe that the language is now acceptable for the review process.

Round 2

Reviewer 3 Report

Comments and Suggestions for Authors

The submitted paper, in the current revised version, focuses on irradiated Poly Vinyl Alcohol-Cellulose Nanofiber Hydrogels samples, with regard to their preparation, material mechanical characterization and employment in artificial joint cartilage tissues.

The revised manuscript barely accounts for the reviewers’ comments. In particular, the Section number 2, devoted to Materials and Methods, shall provide, by text, data, possibly equations, tables and figures a suitable description of the materials and the methods, in order to ensure the scientific soundness and reproducibility of the results.

Author Response

Comments 1: The revised manuscript barely accounts for the reviewers’ comments. In particular, the Section number 2, devoted to Materials and Methods, shall provide, by text, data, possibly equations, tables and figures a suitable description of the materials and the methods, in order to ensure the scientific soundness and reproducibility of the results.

Response 1: Thank you for your valuable comments on our manuscript. We have carefully revised the Materials and Methods section according to your suggestions. We have clarified the experimental conditions and included sample tables. The revised version has been attached for your review. We would appreciate it if you could take a look and let us know if there is anything else we can do to improve the manuscript.
